# The spatial layout of antagonistic brain regions is explicable based on geometric principles
Robert Leech [1] ✉, Rodrigo M. Braga [2], David Haydock [3], Nicholas Vowles[1], Elizabeth Jefferies [4], Boris Bernhardt [5], Federico Turkheimer [1], Francesco Alberti [6], Daniel Margulies [6], Oliver Sherwood[1], Emily JH Jones [7], Jonathan Smallwood [8,9] & František Váša [1,9]

Brain activity emerges in a dynamic landscape of regional increases and decreases that span the cortex. Increases in activity during a cognitive task are often assumed to reflect the processing of task-relevant information, while reductions can be interpreted as suppression of irrelevant activity to facilitate task goals. Here, we explore the relationship between task-induced increases and decreases in activity from a geometric perspective. Using a technique known as kriging, developed in earth sciences, we examined whether the spatial organisation of brain regions showing positive activity could be predicted based on the spatial layout of regions showing activity decreases (and vice versa). Consistent with this hypothesis we established the spatial distribution of regions showing reductions in activity could predict (i) regions showing task-relevant increases in activity in both groups of humans and single individuals; (ii) patterns of neural activity captured by calcium imaging in mice; and, (iii) showed a high degree of generalisability across task contexts. Our analysis, therefore, establishes that antagonistic relationships between brain regions are topographically determined, a spatial analog for the well documented anti-correlation between brain systems over time.

One of the most well-established features of brain activity is a landscape of regional increases and decreases in activity that emerge in humans and other mammals. For example, a distributed set of regions show systematic decreases in activity when participants engage in a range of complex tasks[1]. These regions are known as the default mode network, and have been hypothesised to play a role in cognition and behaviour linked to memory and other internally focused cognitive processes[2]. At the same time, a different set of regions show a tendency to increase activity when participants perform tasks with increased cognitive demands, a system which is often referred to as the multiple demand network and is assumed to play an important role in executive control[3].

Functional accounts typically argue that task-positive or task-negative features of brain activity reflect how the brain implements different features of cognition, such as the application of task rules in the case of executive control[3,4] or the use of long-term knowledge to guide cognition and behaviour in the case of the default mode network[2]. In contrast, in our current study, we consider a complementary hypothesis: that regional increases and decreases in brain activity observed during different states are the result of a set of common topographic causes. To investigate this possibility, we examined whether the spatial distribution of regions that show increases in brain activity can be predicted by the regional distribution of regions that show reductions in brain activity (and vice versa).

The importance of physical space as an organising principle has long been recognised as fundamental for brain function. For instance, sensory and motor functions are arranged as topological maps[5] and processing streams[6,7]. However, an increasing number of studies[2,8–12] have highlighted the possibility that geometric constraints may provide a set of general principles that could explain the rich neural dynamics that are observed across the cerebral cortex and would therefore help constrain how these neural processes support different cognitive functions.

These geometric perspectives suggest that the specific spatial patterns of activation and deactivation we observe with neuroimaging reflect the

[1]Institute of Psychiatry, Psychology & Neuroscience, King's College London, London, UK. [2]Neurology Department, Northwestern University, Chicago, IL, USA. [3]Division of Psychology and Language Sciences, University College London, London, UK. [4]Department of Psychology, University of York, York, UK. [5]Montreal Neurological Institute-Hospital, McGill University, Montreal, QC, Canada. [6]Integrative Neuroscience and Cognition Center, University of Paris, Paris, France. [7]Centre for Brain & Cognitive Development, Birkbeck, University of London, London, UK. [8]Department of Psychology, Queens University, Kingston, ON, Canada. [9]These authors contributed equally: Jonathan Smallwood, František Váša. ✉e-mail: robert.leech@kcl.ac.uk

operation of topographical constraints that relate to how the brain functions as a system. An analogy can be made with landscapes, where the pattern of elevation on the Earth's surface reflects processes that are distributed in space (e.g., plate tectonics, volcanic activity, glaciation, and weathering). For many of these phenomena, the peaks and valleys observed within a specific landscape are a consequence of a small number of geological processes that lead to predictable changes in elevation over space and time that are common across landscapes located in different parts of the globe. Although the spatial processes acting on the cortex are likely to be fundamentally different, we hypothesised that the relative increases and decreases in neural activity that emerge during specific states may, in part, be determined by the operation of neural topographical principles. Such a view of brain function would be supported by evidence that the pattern of regional increases in activity can be predicted by the distribution of regions that show decreases in activity (and vice versa).

In order to determine whether common topographical principles explain both increases and decreases in brain activity during tasks, we used an approach that is commonly used to infer topographical features in geology and other earth sciences. It is well established that because processes like glaciation shape both the peaks of mountains and the valley floors. Consequently, physical descriptions of the shape of a mountain peak contains information that also describes the shape of the valley floor (and vice versa). In a geological context, the relationship between peaks in valleys can be captured by a spatial regression technique called Kriging[13] that has been applied in other domains such as ecology, geography, and climate sciences. Kriging has recently been extended to work with large datasets of tens of thousands of observations and accommodate spatial heterogeneity (i.e., the possibility that underlying influences on a landscape's topography can vary across space), making it a viable technique for facilitating its use in a neuroimaging context[14].

In our study we apply Kriging (Fig. 1, top) to the distribution of brain activity observed in a range of different situations and using different imaging techniques (fMRI and calcium imaging). Specifically, we examined whether the spatial distribution of vertices that show reductions in activity (i.e., vertices on the cortical surface that show a relative deactivation) are able to predict the distribution of vertices that show activity increases (i.e., vertices with a relative increase in activity) and vice versa (Fig. 1, bottom). Our findings highlight that this can be done with remarkable accuracy indicating that: (i) many of the observable features of task-positive brain activity are spatially linked to reductions in activity (and vice versa); (ii) that this is not tied to a specific modality of brain imaging and (iii) while there are unique task-positive activity patterns arising from the spatial configuration of task deactivation for each task there are also important similarities seen when human perform different cognitive tasks, explaining why often different tasks can have similar neural profiles. Together, these analyses establish that the task-positive and task-negative dimensions of brain activity are at least partly an emergent property of cortical geometry. More generally, they

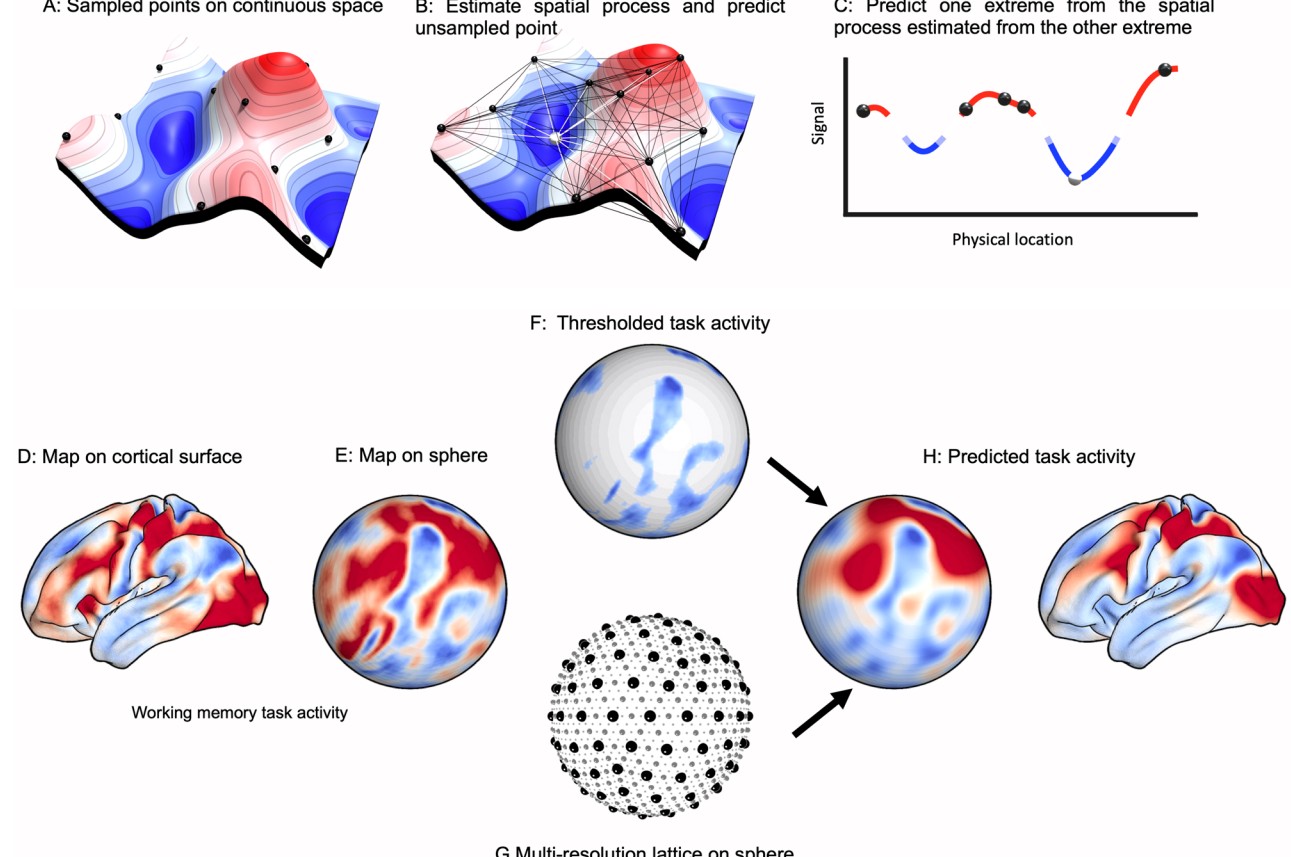

**Fig. 1 | The spatial regression approach (Kriging).** Top, schematic of the spatial regression approach: **A** an illustration of a signal (elevation and color) in a 2D physical space, with black spheres indicating sampled points where the signal has been measured. **B** Distances and similarities of the signal between sampled points (black lines) can be used to approximate the spatial process of the continuous signal. This spatial process can be used to predict unsampled points (white sphere) based only on distance to sampled points (white lines). **C** 2D cut through, showing using samples (black points) from one extreme to predict the other extreme (white point).

Bottom, application of the spatial regression approach to the cortical surface. **D** A cortical surface map is identified, for example, activations from a cognitive task. **E** The map is projected onto a sphere, and **F** thresholded to retain a subset of regions; here, those that show deactivation for the chosen task. Next, **G** a multiresolution regular lattice is created on the sphere, to estimate spatial covariance and **H** performs spatial prediction of the whole-brain map, including in particular the out-of-mask (task-positive) regions. Finally, these can be compared to the true out-of-mask values (i.e., observed task-positive activity).

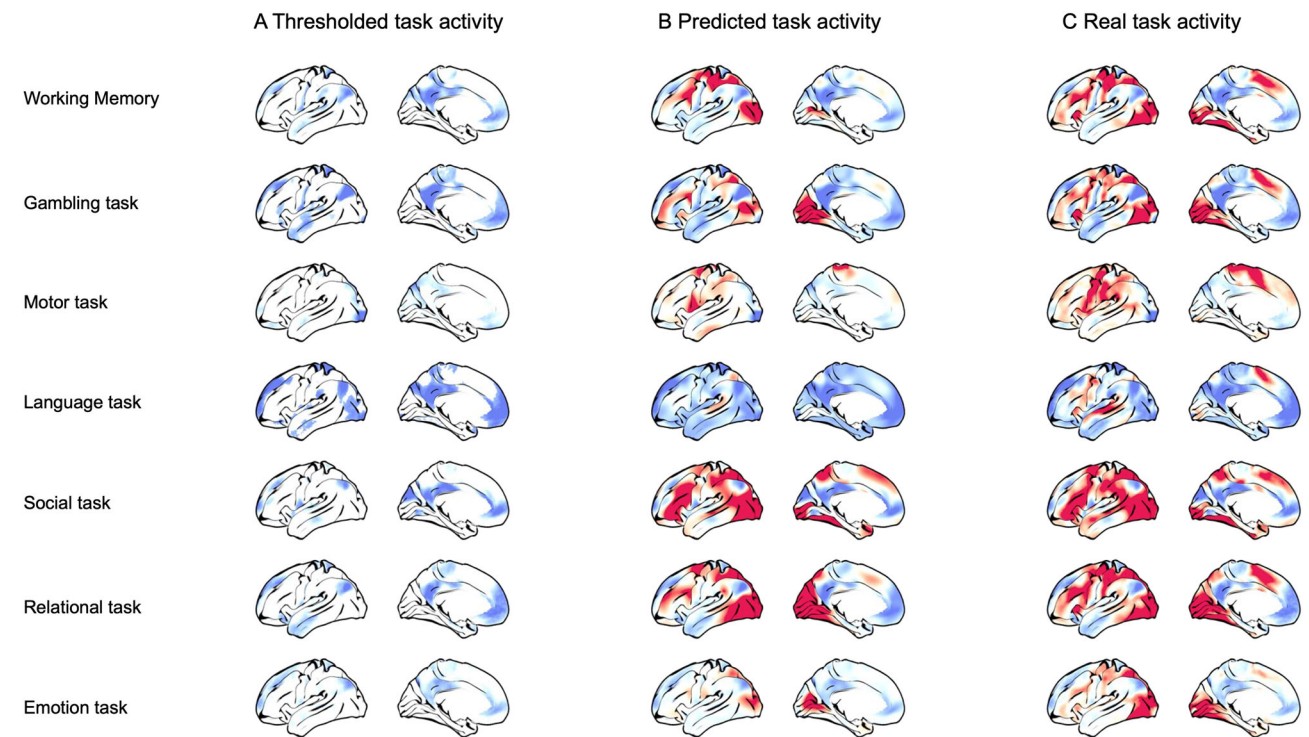

**Fig. 2 | Spatial prediction from task-negative activity patterns. A** Vertices with the lowest (<25%) group-average task activation were used to form a predictor mask. **B** Spatial regression was used to predict task activity at all remaining vertices, and compared to **C** the true pattern of activity.

underscore the importance of mapping both regional increases and decreases in activity in theoretical approaches to the brain basis of cognition.

## Results

We first demonstrate the utility of kriging as a technique for explaining the spatial distribution of increases and decreases in brain activity using group average BOLD activity from the Human Connectome Project[15]. We took a single group-averaged contrast z-score map (e.g., 0-Back relative to baseline) for each of the seven cognitive tasks (see Fig. 2). Each task map was thresholded to retain the lowest 25% of cortical vertices (i.e., most negative values; excluding the medial wall), and the resulting mask was used as the input for spatial prediction. We performed Lattice Kriging by projecting the masked input data onto a multiresolution spherical lattice and using the pattern of changes in activity as a function of distance to approximate the underlying spatial process and, based on this, predict all vertices on the cortical surface (except the medial wall). The model was estimated separately for each task, without prior training on other datasets; as such, each separate model was only provided with the thresholded spatial distribution of activity decreases for that specific task. In this analysis, predictions are entirely based on regions showing decreases in activity and we assess whether this is able to predict the distribution of positive activity in the unseen vertices. Consistent with our hypothesis that a set of common topographical principles explains both increases and decreases in brain activity during tasks, we (Fig. 2A) generated a set of predictions for each task (Fig. 2B), which showed reasonable correspondence to the "true" task patterns (Fig. 2C) (see Supplementary Fig. 1 for the inverse predictions, i.e., predicting task negative vertices from super-threshold task-positive vertices).

We quantified the relationship between predicted and true increases in brain activity in the out-of-mask vertices (Fig. 3A). Spearman's correlations between predicted and empirical activity ranged from $\rho = 0.57$ (for the Emotion task) to $\rho = 0.77$ (for the theory of mind task). The presence of autocorrelation unrelated to task activity (e.g., resulting from thermal noise, registration error between subjects, etc.) means that some positive correlation in out-of-mask prediction is to be expected by chance. Therefore, we also calculated the correlation between true and predicted activity within a restricted set of vertices (Fig. 3B). We restricted the correlation between real and predicted activity to vertices with task-positive activity (i.e., $z$-score $> 0$), excluding any vertices outside of the predictor mask whose activity was negative ($z < 0$). This restricted analysis still showed positive correlations between real and predicted activity (range of Spearman's $\rho$ across tasks = 0.34–0.8). This demonstrated that the model does not just predict the *location* of vertices with positive BOLD activity (or assign all vertices a given distance from the mask as positive), but also predicts the *variability* in activity across these task-positive vertices. Supplementary Fig. 2 shows results using higher and lower thresholds to make the mask. Overall predictive accuracy was somewhat dependent on the level of the threshold (i.e., the amount of vertices included in the mask) as expected; however, the relative predictive performance across the different tasks was broadly consistent across all three threshold levels.

In Fig. 3C, we also observe the similarity in out-of-mask predictive performance (derived from each task-negative pattern of activity) across each of the seven tasks. This illustrates that for the majority of the tasks (with the exception of the motor task), the predicted solution is well-aligned across tasks (although it is always strongest for prediction of its own out-of-mask activity). This indicates a mixture of task-general and task-specific aspects of the task-negative spatial configuration. An informative summary measure is the overlap in the number of tasks predicted to have positive activity at each vertex (Fig. 3D, E), regions that tend to show good spatial agreement between predicted and real data.

In Fig. 3D, E, we observed evidence of both a shared pattern of task-positive activity predictable from task-negative activity for the majority of tasks, as well as elements that are unique to each task.

To further evaluate the task-specific component of the spatial prediction, we next directly compared predictive performance across pairs of tasks (Fig. 4, top). We generated a binary mask based on the intersection of a pair of (previously generated) task-specific masks. We used Kriging to reconstruct the underlying cortical activation maps from a subset of task activity which was spatially restricted only to the intersection mask (i.e., we predicted both increases and decreases in brain activity). Finally, we compared

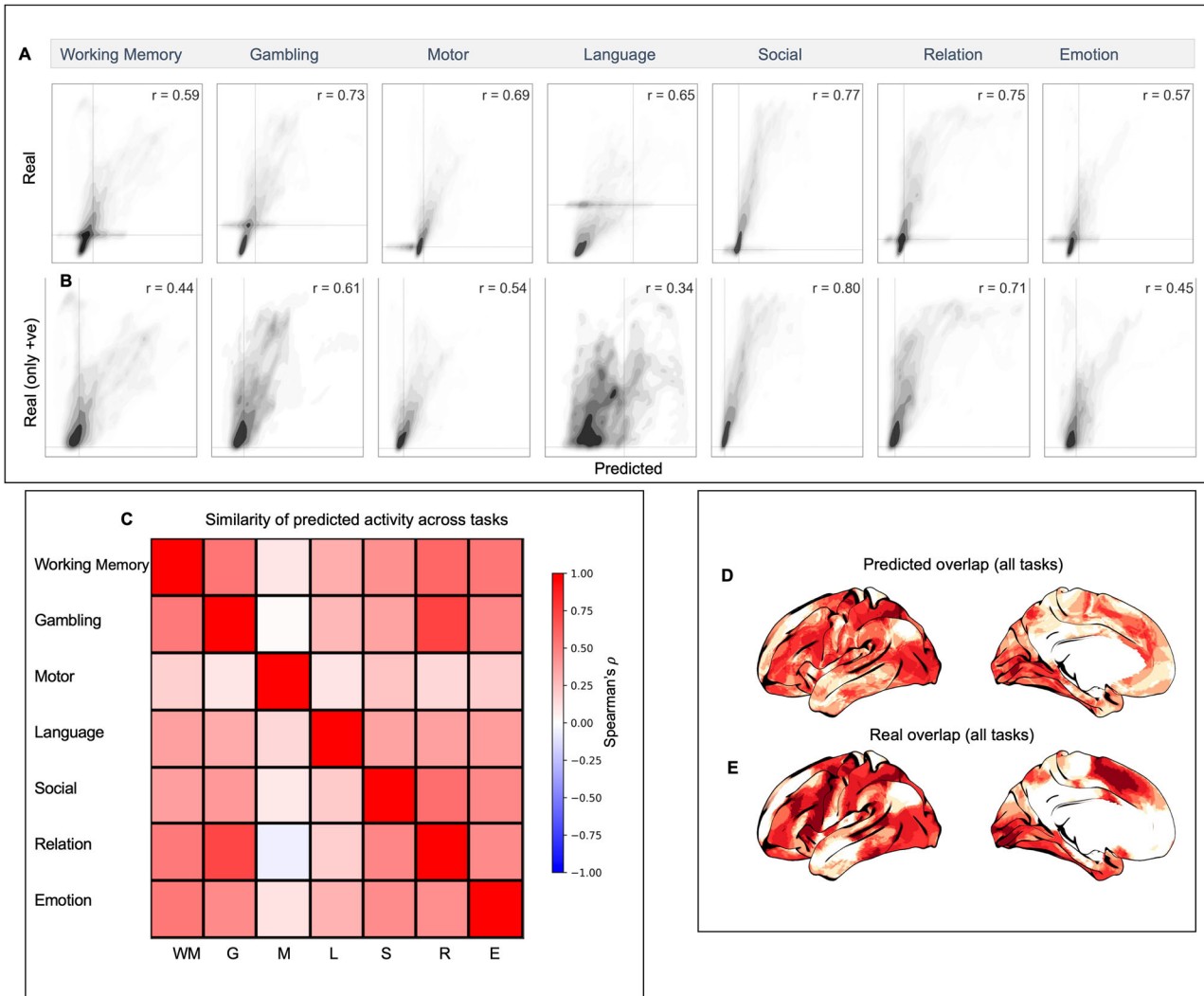

**Fig. 3 | Assessing the agreement between predicted and true patterns of activity.** Left panel, **A** scatter plots of the relationship between true and predicted out-of-mask vertices for each task. **B** The relationship between predicted and real activity restricted only to vertices with a true task-positive BOLD response (i.e., $z > 0$). **C** Similarity across tasks of predicted activity for all out-of-mask vertices. **D** The overlap of positive predicted values across all tasks; **E** the real overlap of positive values across all tasks.

the predicted map to both tested empirical task maps to quantify whether the predicted map is more similar to the corresponding empirical map (whose activation within the intersection mask was used for prediction). We repeated this across all 42 pairs of tasks. This allowed us to systematically compare predictive performance for pairs of tasks based on matching subsets of vertices in the predictor mask. As such, this was a highly constrained test of the hypothesis that the spatial distribution of task activity within the predictor mask allows for meaningful spatial prediction outside of the mask (and is not driven just by the location of task-negative vertices). Figure 4, bottom, shows the comparison of out-of-mask similarity between each pair of tasks (true versus alternative task). Out of all 42 task comparisons, the true task used to generate the prediction had greater predictive performance than the alternative task 39 times (Fig. 4, bottom A). Similar results were obtained when the out-of-mask performance was restricted to either a minimum distance from any vertex in the mask (0.1 radians, ≈3 mm; Fig. 4, bottom B), or only to vertices that had positive activity in the real pattern of activity (Fig. 4, bottom C).

So far, we have shown that the input masks based on regional decreases can predict regions that show positive activity. A complementary perspective on spatial prediction can be obtained by constraining input data to predefined brain regions or networks, and assessing their relative ability to

predict activity in the rest of the brain. Therefore, we also defined masks from seven canonical intrinsic connectivity networks[16] and used these to generate spatial predictions of activations for vertices outside of each mask. Figure 5 shows the correlation between out-of-mask real and predicted activity for each network and each task. Across tasks, the networks covering large primary sensorimotor systems (i.e., visual and somatomotor networks) performed poorly; conversely, association networks (fronto-parietal and default mode networks) showed the best performance. The default mode network (Network 7) is similar in size to the sensorimotor networks (Networks 1 and 2), whereas Network 6 is much smaller, indicating that predictive success is not merely a consequence of network size (i.e., spatial area and number of vertices). The limbic network also performed poorly, although we note that some regions this network have been associated with signal dropout and related issues so may be less biologically meaningful[17].

To assess whether these results arise because of the specific spatial location of each network on the cortical surface, we repeated the analyses following spatial or "spin" permutation of each mask on the sphere[18,19] and using the location of the rotated masks to make predictions. We found similar out-of-mask predictive performance for true networks and for rotated networks (all $p$-values > 0.05, FDR or Bonferroni correction). This

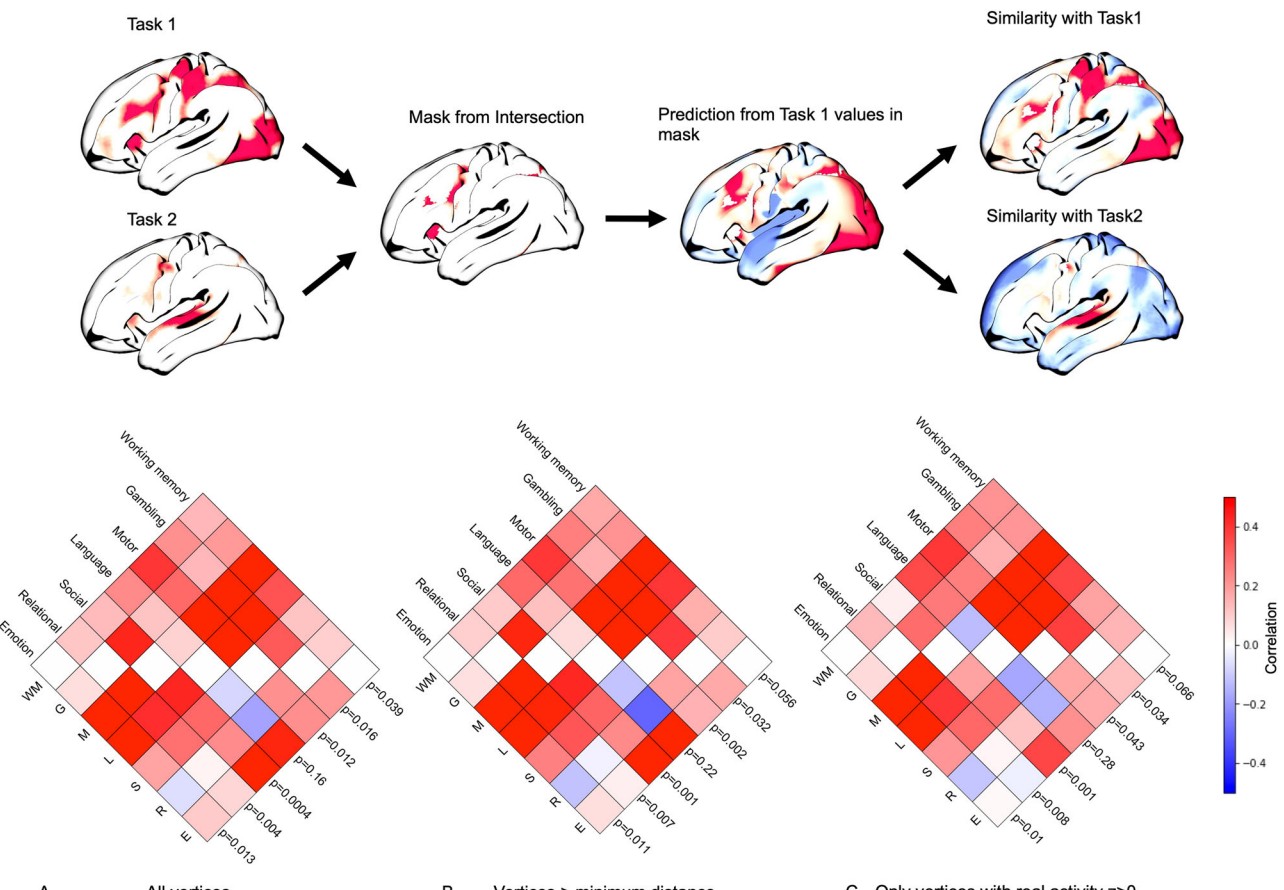

**Fig. 4 | Assessing the task-specificity of spatial predictions.** Top: The approach consists of creating a mask from the intersection of two task-specific masks, predicting activation for one task across the rest of the cortex, and then assessing whether the predicted output is more similar to the (expected) true task than to the alternative task. Bottom: Pairwise similarity in true versus alternative task performance, based on: **A** All out-of-mask vertices; **B** Out-of-mask vertices further than 0.1 radians from any vertex in the predictor mask; **C** Out-of-mask vertices with empirical activity $z > 0$.

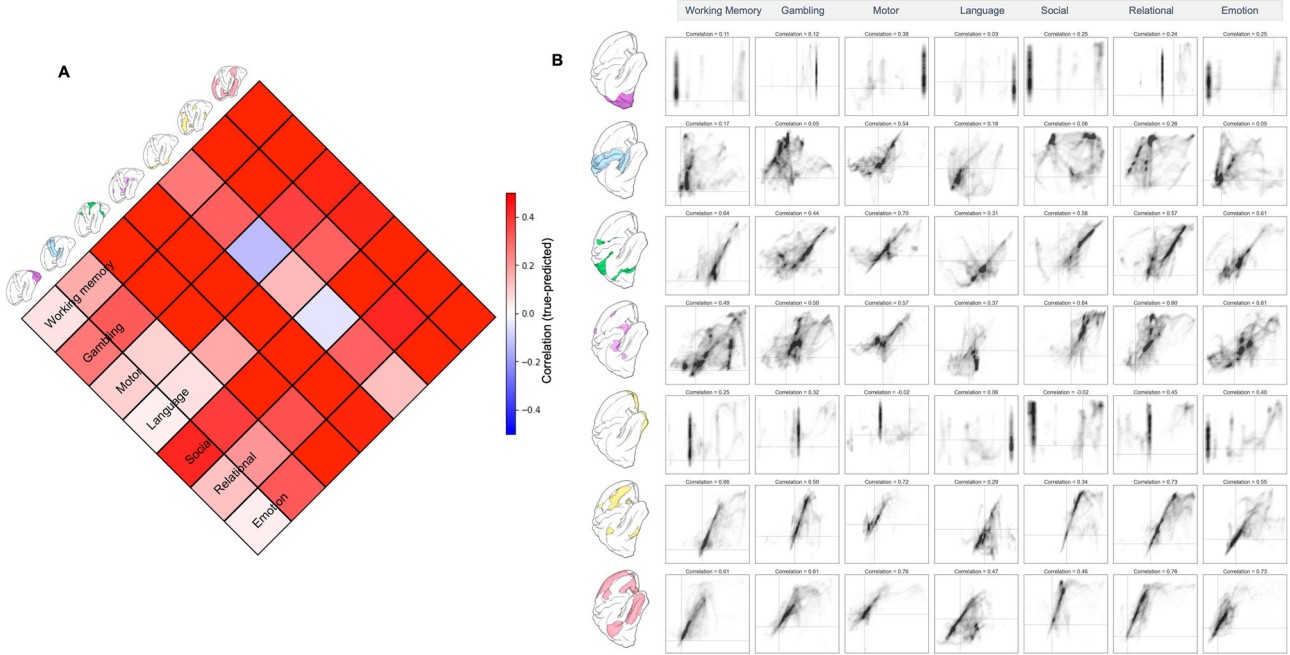

**Fig. 5 | Predicting task activation from masks defined by intrinsic connectivity networks. A** The similarity matrix (correlation values) between real and predicted out-of-mask activity profiles; **B** Scatter plots of the out-of-mask predictions compared to activity for each network and each task.

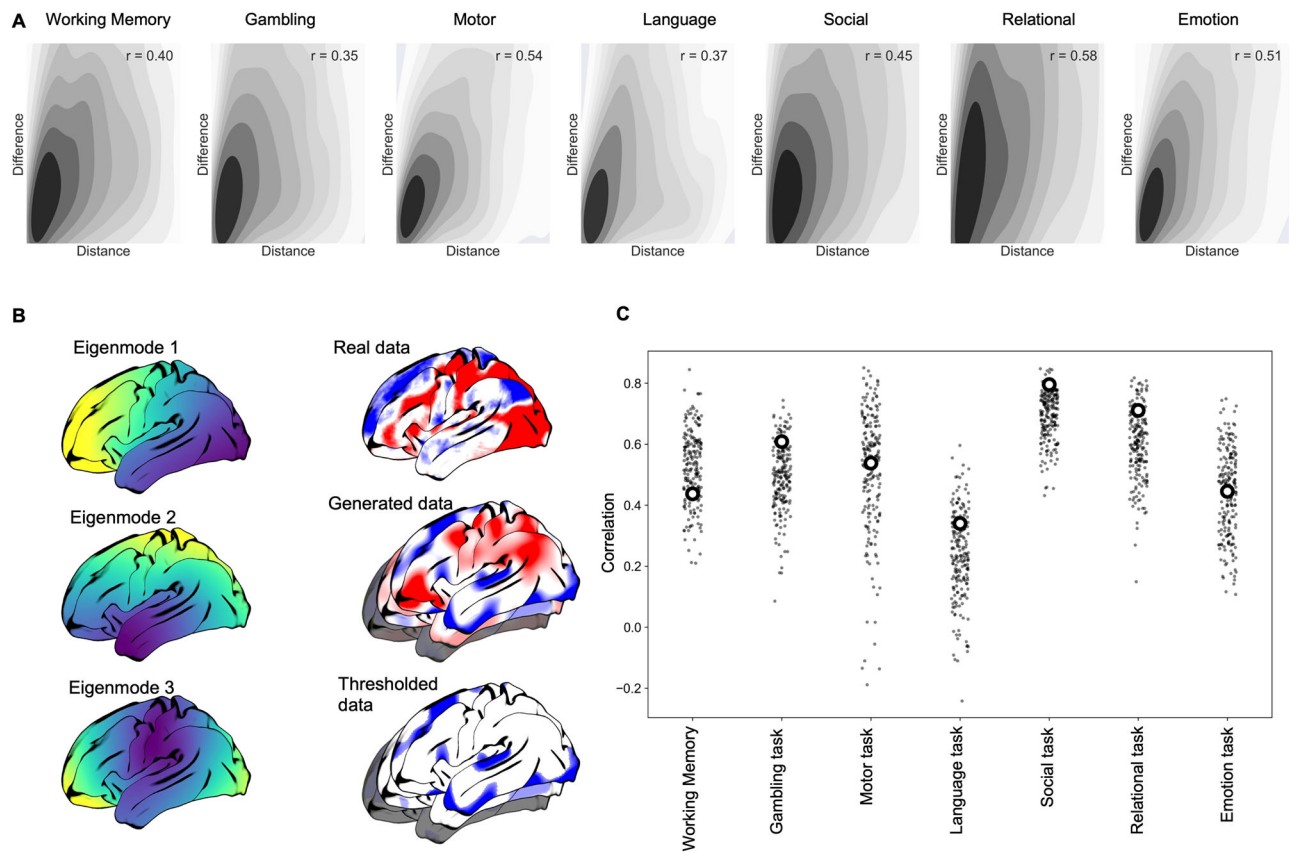

**Fig. 6 | Statistical features underlying spatial predictions.** Upper panel: **A** the relationship between distance (great circle distance) and predictive accuracy for each of the tasks. Lower panel, spatial prediction using randomized task activation maps: **B** illustration of the first three Eigenmodes generated using Eigenstrapping[20]. These were used to generate 200 randomized versions of each task map while approximately preserving the spatial autocorrelation; these surrogate maps were then thresholded (<25% of vertices) and used to spatially predict out-of-mask vertices. **C** The similarity (Spearman's ρ) between true and predicted out-of-mask vertices for the original task map (diamonds) and the randomized maps (dots).

indicates that the superior predictive performance of heteromodal (rather than unimodal) networks is due to their spatially distributed nature, rather than the specific location of their sub-regions.

We next demonstrate that the prediction of variation in task-positive activity from the task-negative troughs of activity is a result of the spatial autocorrelation in the data. In Fig. 6A, we observe that the accuracy of the prediction is related to the minimum distance along the cortex of each vertex from the task negative mask for all seven tasks, with accuracy reducing at longer distances. Second, we generate a set of surrogate maps which match the spatial autocorrelation for each task but randomize the specific locations of peaks and trough using the recently proposed Eigenstrapping method[20] (Fig. 6B). We observe that the predictive accuracy of the true task map is similar to the accuracy observed for 200 surrogate maps, for all tasks (all *p*-values > *0.065*, uncorrected for multiple comparisons). This indicates that the spatial regression is taking advantage of spatial autocorrelation structure in the data to make successful predictions.

So far, we have established that increases in brain activity in humans while they perform tasks, as assessed by fMRI, can be predicted based on the spatial distribution of regions that show decreases in activity. Next we explore two alternative accounts of our data that emerge from contemporary views on the validity of fMRI as a tool for mapping human brain activity: (i) the blurring of signals caused by group averaging and (ii) the lack of biological reality in the fMRI signal.

It is often argued that the method of inter-subject averaging blurs regions with distinct functional profiles so that the time series loses its biological meaning. To address this possibility, we repeated the spatial prediction analysis for three tasks from 10 heavily sampled individual participants from the Midnight Scan Club[21]. Similarly to the group analysis, for each individual, a mask formed from the lowest quartile of vertices was used to predict individual activity for vertices across the cortex. As with the group analysis, the predicted activity was generally similar to real activity, both at the individual level and when averaging the results of individualized predictions (Fig. 7A, B). We also repeated the pairwise comparisons of different tasks (Fig. 4, top) at the individual level, by predicting from the mask of the intersection of the lowest quartile of vertices for each pair of tasks. As with the HCP data, we observed greater out-of-mask prediction for true tasks than the alternative tasks (Fig. 7C). For 7/10 subjects, prediction was superior for true tasks for all 6 task pairs; for the remaining three subjects, prediction for the true task was superior for 5/6 task pairs. In other words, prediction was superior for the true task for 57/60 task pairs across individual subjects (Fig. 7D). This analysis rules out the possibility that our ability to predict increases in brain activity based on the spatial configuration of task-negative activity is an artifact of group averaging.

Until now, we have considered BOLD responses calculated from human fMRI; however, fMRI is often argued to lack the biological reality that is possible with more direct metrics of neural functions, such as intercranial recordings or calcium imaging. Since our analysis depends on the topography of the cortex, recordings of single regions are unsuitable to test our spatial hypothesis. Instead, we investigated whether this approach generalizes to a more direct metric of neural activity: calcium imaging recordings from visual cortex while a mouse watched a movie (Fig. 8). For individual time points, we used the top 25% of pixels to form a mask and used spatial prediction to make out-of-mask predictions (Kriging on a 2D lattice covering the visual cortex). Note that, unlike the human fMRI

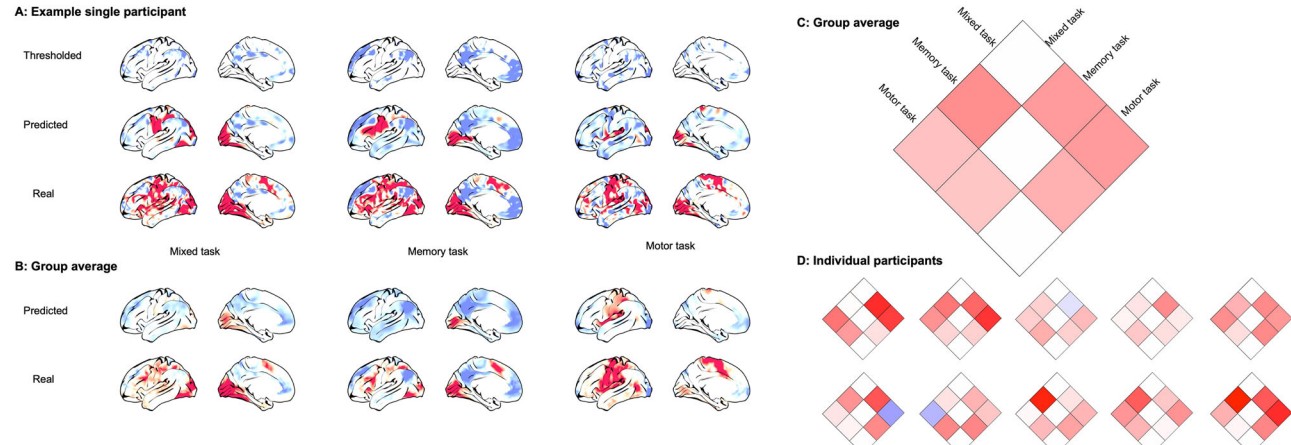

**Fig. 7 | Individual participant analyses. A** The thresholded activity pattern, corresponding prediction and underlying real activity pattern for the three tasks for an illustrative participant projected on an average surface. **B** Averaging the individualized predictions and real activity across participants. **C** Out-of-mask pairwise similarity (correlation) comparing each task (i.e., the predicted compared to true activity for one task compared to the other tasks). **D** The same as C but for each individual participant.

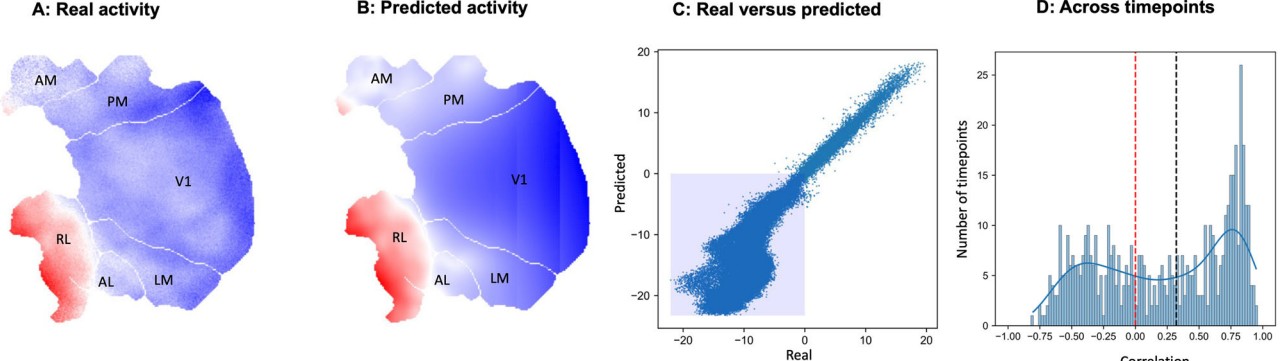

**Fig. 8 | Spatial prediction in a mouse. A** Real activity for a single time point. Labels are: primary visual cortex (V1), lateromedial area (LM), anterolateral area (AL), rostrolateral area (RL), anteromedial area (AM), and posteromedial area (PM). **B** Predicted activity from a mask of the top 25% of pixels. **C** Real versus predicted activity in out-of-mask pixels (shaded area are negative values—the mask was defined by extreme positive values). **D** Top right, distribution of Spearman rank correlations between out-of-mask (>25 pixel distance) real and predicted activity across time points. The black dashed line is the mean correlation.

analyses, which were calculated on contrast maps, this analysis used time series data to predict task-related reductions or increases in activity over time. As with the fMRI data, we observed that long-distance out-of-mask spatial predictions of the pattern of deactivating pixels could be calculated based only on the spatial pattern within the extreme of the distribution. Specifically, we observed positive correlations between real and predicted out-of-mask pixels for 420/540 (78%) time-points, and a mean Spearman correlation of $\rho = 0.62$. This result held even when restricting the out-of-mask prediction to a minimum distance of 25 pixels from the input mask (332/540, 61%, $\rho = 0.31$) (Fig. 8D). Our final analysis, therefore, shows that patterns of activity increases in brain activity can be estimated based on geometry from measures of brain activity that are traditionally seen as more biologically plausible than fMRI.

## Discussion

In this study, we established that the spatial configuration of both increases and decreases in cortical task activity is the result of a common topographic principle. Specifically, across multiple tasks—and for both group averaged and individual data—we found that the spatial organisation of extreme task-negative activity can be used to predict task-positive activity patterns (and vice versa). We could predict not just the spatial location of task-positive activity, but the magnitude of task-positive activity values in a task-specific way. Further, by using intrinsic connectivity networks at rest, we show that

networks with different spatial characteristics differ for predicting cortex-wide activity and that this likely reflects the networks' spatial distribution, rather than its spatial location. Finally, we show in calcium imaging data across mouse visual cortex, that these spatial dependencies are not only restricted to humans or to BOLD functional MRI data.

Our findings indicate that task-positive and task-negative patterns of brain activity, which may be associated with different cognitive processes, are spatially linked and are likely the result of set of topographical principles that influence how neural processes unfold over space and time. Our results, therefore, can be considered as a spatial analogue of the finding that some brain networks are temporally anti-correlated[22]; our analysis shows that observed activity patterns are such that neural activity is also anti-correlated over space. These spatial constraints are in many ways not surprising given the geometric organisation underlying both the distribution of white matter connections and the preponderance of local connectivity across the cortex[23]. Moreover, they are also predicted by approaches that emphasise how topographic features influence the function of specific brain networks (for example, the default mode network[11], or the general topography of observed brain activity based on descriptions of its structure[12].

From a cognitive perspective, one puzzling feature of the task-positive and task-negative features of brain activity is the broad range of situations in which it is important[3]. For example, the multiple demand network, as the name implies, shows a pattern of increased activity across a range of

different tasks, each of which differs in superficial details (for example, they may differ on the type of stimulus, the rate of stimulus presentation, and features of task structure). Likewise, regions in the default mode network, such as the posterior cingulate cortex, show reductions in activity in many of the same sorts of tasks that activate during the multiple demand network, and greater activity in situations in tasks that all share a reliance on memory (e.g., semantic knowledge, episodic memory or social cognition)[2,24]. At a cortical level, therefore, the task-positive/task-negative axis of brain function describes spatial patterns of brain activity that occur across a range of specific situations and that may share abstract cognitive features (such as a role of executive control), but also differ in superficial features of the situation in which they are observed (e.g., the Stroop task is not the same as a working memory task). Although the generality of the increases and decreases in brain activity are now well documented, they beg a specific question: what features of cognition are encoded by features of brain activity that are common to tasks that differ in the specific features of cognition and behaviour that they depend upon? In this regard, our findings are important because they explain that this persistent spatial motif may occur across many superficially different cognitive settings because of the impact of one or more topographical features that shape how brain activity varies across the cortex. In other words, if at least a part of the brain activity pattern we observe during tasks is a result of spatial phenomena which are derived from geometric constraints, then consistent patterns of deactivation and activation should be expected in tasks which may be different in specific features of cognition or behaviour. This is analogous to the way that a small set of common geological processes can explain basic features of the landscape of mountain ranges, even if they occur in different continents[25].

Our analyses establish links between the spatial distribution of increases and decreases in brain activity that can be parsimoniously explained by assuming that a set of common spatial principles governs both positive and negative changes in brain activity observed during tasks. Nonetheless, our analysis leaves open several important questions. On its own, techniques such as Kriging will not directly elucidate the specific cortical mechanisms underlying the spatial processes; instead, they reveal the dependency structures between regional patterns of brain activity, even when these are not obvious. These spatial dependencies act as constraints that should be incorporated into theoretical cognitive neuroscientific accounts (e.g., task-positive and task-negative systems should not be treated as fully independent systems when ascribing cognitive functions). Further, they can be used to guide the application of more mechanistic, generative models, especially in the context of cognitive tasks. For example, recently, Pang and colleagues[12] argued that the neural activity observed during fMRI can be explained by a process in which neural activity is an emergent property of the mechanisms which shape the cortex, arguing that observed brain activity is the result of resonance of spatial features of brain organisation that can be explained by neural field theory. It is possible, therefore, that the spatial phenomenon described in our analysis may be explicable in similar terms.

Methodologically, the spatial regression method used here, is applied to a single input brain map, without prior training on a separate dataset (as is common in supervised machine learning). While we only applied this method to functional imaging data, including human fMRI and mouse calcium imaging, this approach can be applied to a wider range of cortical maps, across imaging modalities and species. For example, a relevant candidate for future work are maps of human anatomical, electrophysiological, and genetic organisation[26]. Future application of these and related methods (e.g.[27].) to a wider range of maps is likely to further enhance our understanding of the spatial relationships between distributed cortical systems.

Future developments also need to focus on the spatial properties of the surface mesh used for spatial regression. Here, we performed the analyses on an approximately spherical cortical projection; while computationally efficient, this projection involves distortion from the true sulcal and gyral geometry of the cortex. Performing spatial regression on a more accurate cortical representation (e.g., mid-thickness surface projection) is an active area of ongoing work which has the potential to substantially improve the accuracy of spatial predictions. We also made simplifying assumptions for computational efficiency, such as that the spatial autocorrelation is spatially homogeneous, which is unlikely to be the case[10,27]. Although technically challenging, incorporating spatial heterogeneity into the spatial regression models in the future will also lead to more accurate characterisation of the dependency between antagonistic patterns of brain activity. The choice of preprocessing steps will also affect the character and magnitude of spatial autocorrelation in the dataset (e.g., steps such as global signal regression) and should be considered systematically in future work. Finally, we observed considerable variability in predictive accuracy across different tasks. There are a number of reasons that this may be the case: acquisition differences (e.g., for the HCP dataset, the FMRI tasks varied in length from 2 to 5 min) and task designs differences (e.g., length of rest periods) both substantially impact BOLD signal sensitivity; equally, the cognitive functions involved in the tasks as well as more general aspects of e.g., arousal vary across the tasks. Future work with more balanced task designs and systematic manipulations will be necessary to allow us to disambiguate whether variation in the strength of spatial dependency between antagonistic networks is a result of cognitive differences or a statistical artefact of task design.

The evidence that geometry constrains the observed patterns of brain activity during tasks has implications for interpreting the links between brain activity, on the one hand, and cognition and behaviour, on the other. For example, it is often standard practice to make inferences about a region's function based on observed increases in functional activity within a specific task. Our data suggests that these inferences could also take into account regions that show reductions in activity, since in many situations the increases in activity contain information about the task context that is also contained in the pattern of reductions. More generally, computational models linking brain geometry, activity, and behavior are necessary to understand the cognitive implications of these geometric constraints. For example, the balance of task evoked activation and deactivation have been considered as spatial homeostatic processes[28–30] that can facilitate richer behavioral dynamics in a reservoir computing model[30]; incorporating geometric dependencies between task-positive and negative distributions of activity in such computational models may allow for a more mechanistic understanding of the functional role of the spatial dependencies observed here. More generally, our findings show that fundamental functional roles of the cortex (especially at a meso- and macro-scopic scale) are poorly understood; future work will need to more directly consider the implications of the cortex as a physically embedded spatial system.

## Methods
### Data
**Group average task data.** We used data from the seven group-average task maps were taken from the WU-Minn HCP 900 subjects data release. Preprocessing was performed by the researchers within the HCP, no additional preprocessing was performed. The data was A single contrast map for each task was used, as follows: the working memory task "0-back" contrast; the gambling task "reward" contrast; the motor task, "average" contrast; the language task "math" contrast; the social task "theory-of-mind" contrast; the relationship task "match" contrast; and the emotion task "faces contrast. Full details of the pre-processing pipeline and creation of the group average maps can be found at the HCP900 release notes[15].

Task FMRI data was first pre-processed according to the minimal pre-processing pipeline prior to projection into native and then common surface space; this involved: gradient distortion correction, motion correction (using FSL's FLIRT), field map correction, boundary-based registration to the participant's T1w scan and projection onto the participant's cortical surface. Cortical surface construction was performed using Freesurfer followed by the MSMAll surface registration pipeline. The task FMRI data was projected onto the fs_LR 32k mesh using the standard HCP fMRISurface pipeline. This involved mapping the voxels in the cortical ribbon onto each participant's cortical surface and transforming them based onto the fs_LR 32k surface mesh. Minimal 2 mm surface-based spatial smoothing was then

applied to the timeseries data. The pipelines are available at https://github.com/Washington-University/HCPpipelines.

To derive BOLD contrast maps, predictor task time courses were convolved with a double gamma hemodynamic response function, temporal derivative terms were calculated for each predictor; the time courses were high-pass filtered with a 200 s cutoff. FSL's FILM was used to correct of temporal autocorrelation. Fixed-effects analyses were performed to combine individual runs and FSL's FLAME mixed effect model was used to derive group level z-statistic task contrasts surface maps. Because the data contains related subjects which was not modelled, associated p-values are not valid and were not used in subsequent analyses.

The xyz-coordinates for each BOLD contrast map were extracted from both the 32k Multimodal Surface Matching[31] mid-thickness projection (for display purposes), and from the spherical projection (for spatial regression and prediction).

**Individual participant data.** We used individual task contrast maps from the Midnight Scan Club dataset[21]. We used the preprocessed data from the ten participants' contrast maps for three tasks (Mixed, Memory and Motor), resampled into 32k vertex atlas; full details of the preprocessing are available at OpenNeuro ds000224.

**Calcium-imaging data.** Preprocessed mouse calcium imaging data was taken from[32]. We used the spatio-temporal data from a single mouse while watching a movie, consisting of $400 \times 400$ pixel images (with a pixel size of $0.01 \text{ mm}^2$) acquired at 540 time-points. Details of data acquisition and preprocessing are available at[32].

**Spatial prediction with Lattice Kriging.** Lattice Kriging (LatticeKrig[14]) is a method for performing spatial prediction using Kriging (a form of Gaussian process regression) with large datasets. It involves building a multi-resolution, compactly supported set of radial basis functions on a regular lattice covering the spatial domain, approximating the covariance structure of the data and allowing for spatial predictions. The sparse basis function decomposition of the spatial covariance in Lattice Kriging guarantees a positive semi-definite covariance matrix (not possible for standard Kriging on a whole spherical mesh), allowing for valid spatial prediction models and highly computationally efficient spatial prediction to the whole cortical surface (10,000 s of vertices). Alternate recent feasible approaches, e.g., Fixed Rank Kriging[33] and INLA[34], could also be evaluated in future.

**Human data**

For human data, we use the LatticeKrig implementation with spherical geometry (using the spherical projection of the FSLR 32k cortical atlas). We use the default parameters from the LatticeKrig example on the sphere (LKSphere). This involved three levels of spatial resolution (153, 625, 2523) of Wendland basis functions built on an icosahedron grid by repeatedly subdividing the triangles of an icosahedron into roughly equidistant points; the relative weighting of the three resolutions was set as alpha = [1,0.25,0.01], based on default of LKSphere example from LatticeKrig. Vertex locations in Cartesian space were converted to longitude and latitude and great circle distances were used to calculate distances between vertices and basis function centres and perform spatial prediction. There was no parameter search or model optimization; however, we show predictive performance on out-of-sample vertices for different key parameters in Supplementary Fig. 3 (the number of levels for the multiresolution lattice, the relative importance of the different resolutions, and the correlation range). While there is variability in the results with different parameters (e.g., reducing the number of levels in general reduces predictive performance), the results are broadly consistent, showing similar predictive performance across the different tasks as the original model.

The same analysis approach was taken for both group and individual task functional MRI datasets. Vertex values for each contrast map were sorted (after removing the medial wall) and the bottom 25% of vertices were used to create a mask (these were negative values, corresponding to task-negative evoked responses). The spherical coordinates of vertices (calculated from their 3D-coordinates on the sphere) in the mask, and the corresponding task contrast values, were entered into a Lattice Kriging model and used to predict all out-of-mask vertices (i.e., vertices outside of the mask). Subsequent analyses compared prediction on all out-of-mask vertices to the corresponding true values, as well as only the subset of out-of-mask vertices that had positive task-evoked responses. Similar analyses were repeated with different thresholds to define the masks (i.e., bottom 15% and bottom 35%).

**Task-specificity of spatial predictions**

To assess whether spatial prediction is task-specific, each task was compared pairwise to each other task (e.g., "Task 1" and "Task 2" below). A conjunction mask was created based on the subset of vertices that were in the bottom 25% for both tasks. To assess prediction, a mask of out-of-mask vertices was created for the remaining 75% of vertex values for Task 1. Vertex values (and their locations) within the conjunction mask were then used to predict out-of-mask vertices based separately on Task 1 and Task 2 values. The predictions from both tasks on out-of-mask vertices were then compared to the true values for Task 1, to assess whether predictive performance was higher for the matching task (Task 1) than for the alternative task (Task 2). This created a non-symmetric out-of-mask pairwise similarity matrix between tasks. Two additional restricted out-of-mask masks were created, as a stricter test of predictive performance: (1) only out-of-mask vertices that also had task-positive values (for Task 1); and (2) only out-of-mask vertices that were also at Haversine distance greater than 0.1 radians from the predictor (i.e., input) vertices.

**Effect of intrinsic network architecture on spatial prediction**

A canonical seven-network decomposition of the cortex[16] was used to create masks of vertices. Task contrast values for all vertices (both positive and negative) within each mask were used to predict vertices outside the mask for each of the seven tasks. To assess whether the spatial location of networks influenced predictive performance, the location/orientation of the networks on the brain was shuffled 500 times using random "spin" rotations of the spherical projection of the cortical surface[18,35]. These were then used to spatially predict task contrast values for out-of-mask vertices; this resulted in a null distribution of predictive performance from the rotated masks[19].

**Generating randomized maps with similar spatial autocorrelation**

The Eigenstrapping method[20] was used to generate 200 surrogate cortical maps with similar spatial dependency structure to each of the task contrast maps. We performed a spectral decomposition of the cortical surface to generate 200 geometric eigenmodes. These eigenmodes were then randomly rotated, combined and resampled to create cortical maps with random patterns with approximately matched spatial autocorrelation to the original task maps.

**Mouse data**

A separate lattice kriging model was performed for each of the 540 time points for the calcium imaging data from a single mouse watching a movie. Given that this is spatio-temporal data rather than a contrast map, there is no equivalent to task positive/task negative pixels. Instead, for each time point, from pixels with a non-zero value, the top 25% of pixels were used to create a mask and the corresponding pixel values and all pixel locations were used to predict the response for: (i) out-of-mask pixels; (ii) out-of-mask pixels at a minimum distance of 25 pixels from any pixel within the input mask.

For the mouse data, the cortical surface (composed of 57867 pixels) is imaged as a 2-D plane. Therefore, a 2-D regular lattice was used for Kriging, with multi-resolution basis functions with three levels. See the code for full details.

Performance was assessed by comparing Fisher transformed correlation values between predicted and real values for each time point. Both t-tests and autoregressive models were used to assess whether the average correlation was greater than chance across time points.

## Reporting summary

Further information on research design is available in the Nature Portfolio Reporting Summary linked to this article.

## Data availability

All data included in the present analyses were acquired with informed consent (where appropriate) and are available at the following sources: https://db.humanconnectome.org/ https://openneuro.org/datasets/ds000224 https://nih.figshare.com/collections/Dataset_for_Sit_Goard_Distributed_and_Retinotopically_Asymmetric_Processing_of_Coherent_Motion_in_Mouse_Visual_Cortex/5018363.

## Code availability

Code to repeat the analyses is publicly available at: https://github.com/ActiveNeuroImaging/SpatialAnticorrelations.

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

## Acknowledgements

The authors received support from the Wellcome/EPSRC Centre for Medical Engineering (Ref: WT 203148/Z/16/Z), Simons Foundation (SFG640710) and support from the NIHR Maudsley Biomedical Research Centre and the Data to Early Diagnosis and Precision Medicine Industrial Strategy Challenge Fund, UK Research and Innovation (UKRI).

## Author contributions

R.L., J.S., and F.V. designed the study; R.L., R.B., D.H., N.V., E.J., B.B., F.T., F.A., D.M., O.S., E.J., J.S. and F.V. contributed to the analysis of the data and the manuscript.

## Competing interests

The authors declare no competing interests.
