## [Transparent Peer Review file · Communications Biology]

The spatial layout of antagonistic brain regions is explicable based on geometric principles

Corresponding Author: Professor Robert Leech

Version 0:

Reviewer comments:

Reviewer #1

(Remarks to the Author)

The spatial layout of antagonistic brain regions by Leech et al.

This paper reports the use of a geometrically-informed method for predicting the peaks of cortical surface task fMRI data from knowledge confined to the location of the troughs of activity. To this end, the authors employ a method (“kriging”) developed in earth sciences and apply it through a series of didactic analyses to human fMRI and mouse calcium imaging data. The papers adds to the burgeoning literature on the role of large-scale geometric processes in shaping cortical responses, which now complement the traditional focus on isolated peaks and clusters. The work is well written and the findings presented in a clear pedagogical style. I have some suggestions that may further improve the paper:

1. The authors do a nice job of comparing their work to recent studies on neural field theory (NFT) and geometric eigenmodes (line 424-). NFT is physiologically constrained and anatomically informed, resting on neural responses through an exponential-distance rule of connectivity, as per [12]. Aside from the mentioning of geographical formation in earth sciences, the mechanistic insights afforded by the use of kriging are not clear. Could the authors add some text here – are their in fact any likely common processes at play for large-scale geographical processes and the shaping of cortical activity? If not, that is fine - because it is clearly still an informative demonstration - but this should also best be articulated.
2. What are the statistical properties that allow peaks to be predicted from troughs using kriging? Is it just the spatial correlation of the data, particularly the high power at long spatial wavelength (this is my hunch). This should ideally be tested by generating null data that preserve the spatial autocorrelation but randomize the relative spatial collocation of the troughs and peaks. The spin test will not suffice as I’m sure the authors are aware since the relative collocation of these features is perfectly preserved – just moved around – so the null will not be properly represented. BrainSMASH could be used although it only preserves power at relatively short wavelengths (up to the maximum kernel employed). In my opinion the recent eigenstrapping method [3] would likely suffice since it preserves (approximately) the entire spatial PSD but randomizes other features, such as the relative location of peaks and troughs (i.e. the third order spatial correlations). A “null finding” here would not diminish the work but rather further underline the growing recognition of the role of SA. Rejection of the null would IMO suggest higher order textural properties are informing cortical brain maps (as long as kriging is sensitive to those). For transparency I am an author of this method so the authors are free to motivate and employ another null method.
3. The authors should consider/discuss the role of the GSR and any other mean centring preprocessing step that enforces peaks and troughs to be centred around a zero baseline (this may not be particularly relevant to the method though).
4. The term “out of sample” (e.g. line 183) might be a bit confusing here. I assume it means that the peaks are removed and the kriging method “trained” on troughs then “tested” on peaks in the same map. A lot of people might think out of sample refers to training and a subset of brain maps/subjects and testing on held-out data from other brain maps/subjects.
5. It may not be relevant but the authors should be aware that the cortical surface maps in fsl are not isometric (i.e. vertices are closer together in sulci than gyri). There is a lot of distortion when projecting onto the sphere, where it seems the regression and prediction is performed, line 471 – this might influence a spatially informed method.

Minor:

1. The panels in Figure 3A and 5B are so small as to be hard to really interpret.
2. Figure 4 requires a colorbar (and ideally some thresholding for “statistical significance” although the consistent direction of the subtraction is already informative).
3. Discussion: I don’t see the link to reservoir competing or criticality aside from a very vague metaphor about spatial processes. Even the reference to homeostasis is very vaguely heuristic.

I hope these comments might improve this interesting paper – Michael Breakspear

Reviewer #2

(Remarks to the Author)

This manuscript describes a study that uses spatial analysis of task based fMRI in humans and calcium imaging in mice to argue that the spatial layout of antagonistic brain regions is explicable based on geometric principles. The approach expands upon recent high-profile studies proposing that brain geometry can account for the topography of both spontaneous and task-evoked brain activity in humans (Peng et al, Nature, 2023). In contrast to the Peng study, this manuscript pursues the idea that the topography of brain deactivations can be used to accurately predict that for brain activation and vice versa. Tests of this idea in humans used group-level brain activation maps from multiple tasks in the HCP dataset as well as individual level maps from the midnight scan club.

This manuscript has several strengths. The idea of searching for simple organizing principles that explain classical observations is important and provides a valuable and stimulating counterpoint to conventional theoretical frameworks. The use of multimodal data from different species is great to see. The writing and visualizations are also generally very clear. The Discussion is thought provoking.

I think the work may benefit from further consideration of the following issues however. To my mind, these issues collectively operate to limit interpretability and impact of the work as it is currently presented.

1. The need for a harder test of the phenomenon being examined. In their results, the authors state “Consistent with our hypothesis that a set of common topographical principles explains both increases and decreases in brain activity during tasks we (Figure 2A) generated a set of predictions for each task (Figure 2B), which showed reasonable correspondence to the ‘true’ task patterns (Figure 2C) (see Supplementary Figure 1 for the inverse predictions, i.e., predicting task negative vertices from super-threshold task-positive vertices).”. This is a core observation and interpretation in their paper. However, if you take a mountain range, and sample valley floors including their upward sloping walls - the “missing surface” will almost certainly be peaks. The same hold in reverse if you sample peaks and their slopes down towards the valley floor. As the authors say, the activation map is directly analogous to this mountain range situation - so I find it hard to imaging a result other than what they are reporting. I may be missing something, but it seems almost a necessity. There may however be a way to subset analyses such that correspondence between observed and predicted values is more surprising. For example, rather than taking the extreme negative 25% vertices (and then the extreme positive 25%) to predict the rest, one could estimate the agreement between predicted and observed using graded bins of in- and out-of sample vertices. E.g. If vertices are chopped into deciles by activation, for each task, one would have a 10*10 asymmetric square matrix per task, where vertices in one decile are being used to do the prediction, and the concordance between predicted and observed is shown separately for vertices in all other deciles of activation

2. Showing how regionally variable the predictive ability is and factors that might explain this variability. The author state “The presence of autocorrelation unrelated to task activity (e.g., resulting from thermal noise, registration error between subjects, etc.) means that some positive correlation in out-of-sample prediction is to be expected by chance. Therefore, we also calculated the correlation between true and predicted activity within a restricted set of vertices (Figure 3B).” The smoothness of cortical maps is generally spatially heterogenous. Moreover, one might expect better prediction along valley walls rather than in mountain plateaus. It would therefore be good to see some anatomical maps of how well observed activity is modeled by predicted. For example, you have a local disc of vertices and compare estimated to observed values within this disc then give the concordance value to the central vertex for visualization. Is there spatial variation in how well things are predicted ? Relatedly, it would be good to see a more thorough treatment of whether the accuracy of prediction varies based on the distance of out of sample vertices from nearest within-sample vertex.

3. Specifying and justifying key methodological decisions and showing robustness of findings to arbitrary or principled deviations from these decisions. For example:

- HCP data: How was relatedness between participants dealt with in generating group activation maps? Which specific contrasts ? What were basic components of preprocessing - especially as relates to motion (potential source of spatial correlations)? What was the surface co-registration approach used? These matter and I don't think it's enough to point the reader elsewhere for information.

- Kriging: There are multiple choices in implementation of kriging, and it would be important to show the stability of key findings across different parts of this parameter search space. There are also other alternatives to Kriging for spatial regression (e.g. Gaussian Process Interpolation) and it would also be good to show reproducibility/sensitivity of the finding across these alternatives. It would be good to see how stable results are across different vertex mesh densities. It would also be important to use cut-offs other than 25% (see comment above for more complete treatment of this issue)

4. A general feeling of the manuscript being light in detail (and perhaps even a bit rushed ?) in places. This impression arises in many places.

- In Methods: The work is methodologically complex, but there are several places where the reader is referred to the code or

other papers for details that are importance to have specified in the main work. The details matter, and - not only do they need some brief mention in the text - but we need to know how sensitive findings are the many choices buried in these methods (see above).

- In Figures: Panel 1C does not correspond with 1B (white node to be predicted is in a valley in B, but in a peak in C. There are no network labels or brain map for Yeo-Krienen analyses in Figure 3F. Row text placement is inconsistent across heat maps (e.g. Fig 3C and 4C) and not available for both rows and columns where that would help (e.g. Figure 4C). Scatterplot axes need labeling (real vs, predicted) in Fig 5B.

- In Discussion. There is little consideration of limitation and caveats. Also - the authors observe important variations in predictability that are left entirely undiscussed. What about the observed instances of variable predictability across tasks (Fig3B), as a function of functional network mask (Figure 5), or across individuals (Figure 6D) ?

Version 1:

Reviewer comments:

Reviewer #1

(Remarks to the Author)

Thank you for your thoughtful responses to my prior concerns and in particular for undertaking the additional analyses required to address these points.

Michael Breakspear

Reviewer #2

(Remarks to the Author)

In my first round of comments, I stated ...

"1. The need for a harder test of the phenomenon being examined. In their results, the authors state "Consistent with our hypothesis that a set of common topographical principles explains both increases and decreases in brain activity during tasks we (Figure 2A) generated a set of predictions for each task (Figure 2B), which showed reasonable correspondence to the 'true' task patterns (Figure 2C) (see Supplementary Figure 1 for the inverse predictions, i.e., predicting task negative vertices from super-threshold task-positive vertices)". This is a core observation and interpretation in their paper. However, if you take a mountain range, and sample valley floors including their upward sloping walls - the "missing surface" will almost certainly be peaks. The same hold in reverse if you sample peaks and their slopes down towards the valley floor."

The authors response to this includes the clarifying change to manuscript text which reads ...

"This demonstrated that the model does not just predict the location of vertices with positive BOLD activity (or assign all vertices a given distance from the mask as positive), but also predicts the variability in activity across these task-positive vertices."

However, my point was not simply that one would expect held out positive vertices predicted from a negative mask to be positive rather than negative. The mountain range analogy is that if you know the shape of the valley floor and extrapolate it - the model would not predict flat elevated plateaus outside valley floor, but rather peaks with upward sloping sides. That is - the location and magnitude of activity (altitude) outside the negative mask (valley floor).

I do not see any new analyses presented that address my initial concerns on this point.

In essence do not think it is surprising that you can do a pretty good job of predicting the location and magnitude of one extreme of cortical activation from the other, and that these predictions should show some task specificity. I would expect that. In fact, I am not sure what else one could expect to see given that cortical organization at all levels we've measured is about spatial smooth variations across a sheet.

I am a bit confused as to the disconnect here - which I've not experienced in peer review to date. This makes me think one of two things - either I am not fully understanding what is being done technically or the disconnect is in the null expectations I have as compared to the author's. I am pretty sure I follow the methods, so - with my Editorial hat on - I would suggest finding a third reviewer. Apologies to have hit a bit of a block here and I regret any delays that it may be causing.

I would like to express by gratitude for having been invited to review this work, and my respect for the authors and the review process.

Reviewer #3

(Remarks to the Author)

As instructed in the review invitation, I did not fully critique the entire paper but instead focused only on whether the authors

have adequately addressed the comments of the previous Reviewers (especially Reviewer #2). My assessment is that, in general, the authors have done a great job in addressing most of the comments from the two Reviewers. However, there are some that I believe are not sufficiently addressed in detail. These are itemized below.

1. Re Reviewer 1 Comment 2. It is good to see that the authors have performed additional analysis on null testing. However, the 20 random instances in Figure 6C are not sufficient to make any statistically valid conclusions.

2. Re Reviewer 2 Comment 3 on Kriging. The authors mentioned that they used the LatticeKrig method without the need for parameter search or optimization. However, the LatticeKrig software has intrinsic parameters, such as the number of resolutions of interest, weight that controls the correlation range, and overlap that controls the overlap of the radial basis functions. These could have been set on default values chosen by the software developers. However, I find it important that the authors mention these parameters because the Kriging method is something unfamiliar to researchers in the field, and also perform a thorough robustness check, as the Reviewer suggested, to ensure consistency of results.

3. Re Reviewer 2 Comment 3 on cutoffs. The authors mentioned that changing the cutoffs leads to qualitatively similar results. This is not supported by data in their new Supplementary Figure 2, showing that there is variability in results with respect to the choice of cutoff. This is especially true for results relevant to analysis restricted to task-positive vertices. The authors need to thoroughly check this and discuss it adequately.

Version 2:

Reviewer comments:

Reviewer #3

(Remarks to the Author)

I have no further comments to add.
